# Oxidative Stress Induces a VEGF Autocrine Loop in the Retina: Relevance for Diabetic Retinopathy

**DOI:** 10.3390/cells9061452

**Published:** 2020-06-11

**Authors:** Maria Grazia Rossino, Matteo Lulli, Rosario Amato, Maurizio Cammalleri, Massimo Dal Monte, Giovanni Casini

**Affiliations:** 1Department of Biology, University of Pisa, 56126 Pisa, Italy; mariagrazia.rossino@phd.unipi.it (M.G.R.); rosario.amato@biologia.unipi.it (R.A.); maurizio.cammalleri@unipi.it (M.C.); 2Department of Experimental and Clinical Biomedical Sciences “Mario Serio”, University of Florence, 50134 Florence, Italy; matteo.lulli@unifi.it; 3Interdepartmental Research Center Nutrafood “Nutraceuticals and Food for Health”, University of Pisa, 56124 Pisa, Italy

**Keywords:** Nrf2, VEGFR2, MIO-M1, retinal explant, Müller cell, HIF-1, diabetic retinopathy, conditioned medium

## Abstract

Background: Oxidative stress (OS) plays a central role in diabetic retinopathy (DR), triggering expression and release of vascular endothelial growth factor (VEGF), the increase of which leads to deleterious vascular changes. We tested the hypothesis that OS-stimulated VEGF induces its own expression with an autocrine mechanism. Methods: MIO-M1 cells and ex vivo mouse retinal explants were treated with OS, with exogenous VEGF or with conditioned media (CM) from OS-stressed cultures. Results: Both in MIO-M1 cells and in retinal explants, OS or exogenous VEGF induced a significant increase of *VEGF* mRNA, which was abolished by VEGF receptor 2 (VEGFR-2) inhibition. OS also caused VEGF release. In MIO-M1 cells, CM induced *VEGF* expression, which was abolished by a VEGFR-2 inhibitor. Moreover, the OS-induced increase of *VEGF* mRNA was abolished by a nuclear factor erythroid 2-related factor 2 (Nrf2) blocker, while the effect of exo-VEGF resulted Nrf2-independent. Finally, both the exo-VEGF- and the OS-induced increase of *VEGF* expression were blocked by a hypoxia-inducible factor-1 inhibitor. Conclusions: These results are consistent with the existence of a retinal VEGF autocrine loop triggered by OS. This mechanism may significantly contribute to the maintenance of elevated VEGF levels and therefore it may be of central importance for the onset and development of DR.

## 1. Introduction

Vascular endothelial growth factor (VEGF) is a key factor involved in the pathogenesis of diabetic retinopathy (DR). In particular, prolonged expression and release of VEGF in the retina is responsible for the breakdown of the blood–retina barrier and the proliferation of retinal neovessels causing major visual deficits in DR patients. Current therapies are mainly directed at the suppression of the neovessels with laser photocoagulation or at the sequestration of VEGF with intraocular injections of specific antibodies or traps [1]. In general, current treatment strategies for DR are affected by a variety of limitations and drawbacks [2]. A much better approach for the treatment of DR would be acting upstream to remove, or at least attenuate, the causes of VEGF overexpression. Therefore, an accurate understanding of the mechanisms regulating VEGF in the diabetic retina is needed.

There is evidence suggesting that VEGF is expressed and released in the retina as a neuroprotective factor when adverse conditions threaten neuronal survival [3,4]. In effect, neuroprotective properties of VEGF are well documented [5] and it is conceivable that this growth factor is used by the retina in the context of a neuroprotective strategy in emergency situations where the integrity of retinal neurons is at risk. In particular, oxidative stress (OS) is likely to be a primary causative event in DR [3], and increased OS in retinal models has often been reported to cause VEGF overexpression. Conversely, treatments reducing OS have been found to also reduce VEGF expression [6].

Our hypothesis is that, during diabetes, VEGF, once induced by OS, is progressively upregulated in the retina when adverse conditions persist for long periods, thus leading to the proliferative episodes typical of late stages of DR. One of the mechanisms involved in this prolonged VEGF upregulation may depend on an autocrine loop in which VEGF itself would promote its own expression. This possibility is supported by observations reporting autocrine VEGF binding to its receptor VEGF receptor 2 (VEGFR2) in different cell types to promote cell survival, migration, or differentiation [7,8,9,10,11,12,13,14], or in endothelial cells to induce its own expression [15,16,17,18,19]. Among the various retinal cell types, Müller cells may play an important role in this hypothesized autocrine loop supporting VEGF overexpression in retinal disease. Indeed, these cells are the main producers of VEGF in the retina, express both VEGFR1 and VEGFR2, and have been found to depend on VEGF signaling for their own survival or viability [20,21]. Interestingly, VEGF signaling in Müller cells has been reported to be necessary to preserve the integrity of retinal neurons [22]. OS is known to induce nuclear factor erythroid 2-related factor 2 (Nrf2) activation and nuclear translocation [23], while hypoxia-inducible factor-1 (HIF-1) is known to be involved in the regulation of VEGF expression [24], therefore these factors are also likely to be involved in this mechanism.

In the present investigation, we used both an in vitro and an ex vivo model to test the hypothesis that OS, acting through Nrf2, promotes *VEGF* gene expression and VEGF release in the retina, and that the overexpressed VEGF promotes an autocrine loop, involving VEGFR2 and HIF-1, to induce its own expression. We also considered the possibility that Müller cells may play a primary role in this mechanism.

## 2. Materials and Methods

### 2.1. In Vitro Studies

#### 2.1.1. MIO-M1 Cell Culture

In vitro studies were performed using MIO-M1 cells, kindly provided by Dr. Gloria Astrid Limb (Division of Ocular Biology and Therapeutics, UCL Institute of Ophthalmology, London, UK). MIO-M1 is a spontaneously immortalized human Müller cell line, which retains morphologic features, marker expression and electrophysiological responses of primary isolated Müller cells in culture. MIO-M1 cells were cultured in Dulbecco’s Modified Eagles Medium (DMEM, Lonza, Basel, Switzerland) containing 4.5 g/L glucose supplemented with 10% fetal bovine serum (FBS, Euroclone, Milano, Italy), 100 U/mL Penicillin-Streptomycin (Euroclone), 2 mM LGlutamine (Euroclone) in a humidified incubator at 37 °C in 5% CO_2_. The experiments were performed at 60–80% cell density.

#### 2.1.2. Cell Viability/Proliferation

MIO-M1 cell viability/proliferation was determined using the MTT (3-(4,5-dimethylthiazol-2-yl)-2,5-diphenyltetrazolium bromide) assay. Briefly, after MIO-M1 cells were cultured in 96-well plates overnight, cells were treated as indicated in FBS-free media. Subsequently, 1 mg/mL MTT was added and incubated for further 3 h. After that, an equal volume of dissolution buffer (isopropanol, 4 mM HCl, 0.1% Nonidet P-40) was added to each well for 30 min to dissolve formazan product. Absorbance was measured at 595 nm using the iMark microplate reader (Biorad, Hercules, CA, USA) for the cell viability calculation while the absorbance of the controls was set as 100% of cell viability.
(1)$$\%\;Cell\;Viability=\frac{Sample\;Absorbance}{Control\;Absorbance}\times 100$$

Analyses were executed in three independent experiments, with five experimental replicates for each experimental point.

#### 2.1.3. Pharmacological Treatments

MIO-M1 cells were treated with different concentrations of recombinant human VEGF (Sigma-Aldrich, St. Louis, MO, USA), of H_2_O_2_ (Sigma-Aldrich), or with 0.1 µM Apatinib (Selleck Chemicals, Houston, TX, USA), a VEGFR2 inhibitor. Before the treatments, the growth medium was replaced with FBS-free medium. H_2_O_2_- and VEGF-treated cultures were also treated with 5 µM ML385 (Bio-Techne, Minneapolis, MN, USA), an inhibitor of Nrf2, and with 5 µM acriflavine (ACF, Sigma-Aldrich), a HIF-1 inhibitor. All treatments were performed for 24 h.

#### 2.1.4. Preparation of Conditioned Medium

For the production of conditioned medium (CM), MIO-M1 cells were cultured in 12-well plates overnight (1.5 × 105 cells/well). Subsequently, the cells were washed with PBS and then incubated in 300 µL FBS-free medium containing 400 µM H_2_O_2_ for 24 h. The CM was collected and centrifuged for 5 min at 1200× *g* to remove cell debris. Two hundred fifty µL of supernatant was transferred to fresh MIO-M1 cells cultured in 12-well plates and treated as indicated.

#### 2.1.5. Quantitative Real-Time PCR

Total RNA was extracted (TRI reagent, Sigma-Aldrich), resuspended in RNase-free water and quantified via spectrophotometric analysis (NanoDrop One/One, ThermoFisher Scientific, Waltham, MA, USA). First-strand cDNA was generated from 200 ng of total RNA (Improm II Reverse Transcription System, Promega, Madison, WI, USA). Quantitative real-time PCR (qPCR) was performed using GoTaq qPCR Master Mix (Promega). The qPCR analysis was carried out in triplicate using the CFX96 Real Time PCR Detection System (Bio-Rad Laboratories, Hercules, CA, USA). The primers were designed according to published human cDNA sequences in the GenBank database: *VEGF* 5′-TACCTCCACCATGCCAAGTG-3′ forward and 5′-ATGATTCTGCCCTCCTCCTTC-3′ reverse; β2-microglobulin (*B2M*) 5′-AGTATGCCTGCCGTGTGAAC-3′ forward and 5′-GCGGCATCTTCACAAACCT-3′ reverse. mRNA was quantified with the ΔΔCt method; *VEGF* mRNA levels were normalized to *B2M* mRNA levels as endogenous control.

#### 2.1.6. Enzyme-Linked Immunosorbent Assay (ELISA)

VEGF levels were measured in culture media to evaluate VEGF release using a kit for human VEGF (R&D Systems, Minneapolis, MN, USA). The ELISA plates were evaluated spectrophotometrically (Microplate Reader 680 XR; Bio-Rad Laboratories). All experiments were run in duplicate. After statistical analysis, data from the different experiments were plotted and averaged in the same graph.

#### 2.1.7. Immunofluorescence

MIO-M1 cells grown on µ-Slide 8-well chamber (IBIDI, Gräfelfing, Germany) and treated as indicated, were washed twice with 1 mL of cold PBS, fixed for 20 min in 4% paraformaldehyde in PBS and permeabilized with 0.3% Triton X-100 in PBS for 5 min. This procedure did not alter MIO-M1 cell morphology, as determined with DIC microscopy (Appendix A). Cells were incubated in blocking buffer (5% FBS and 0.3% Triton X-100 in PBS) for 1 h at room temperature. Then, the cells were incubated overnight at 4 °C with an anti-VEGFR2 antibody (ab2349, Abcam, Cambridge, UK; 1:400 dilution) or with an anti-Nrf2 antibody (ab62352, Abcam; 1:500 dilution) and successively with an anti-rabbit secondary antibody conjugated with Alexa-Fluor-488 (Life Technologies, Carlsbad, CA, USA, 1:200 dilution) or with an anti-rabbit secondary antibody conjugated with Cy3 (Sigma-Aldrich, 1:200 dilution) for 1 h at room temperature. After staining of the nuclei with Hoechst 33242 dye (40,6-diamidino-2-phenylindole; ThermoFisher Scientific) and actin filaments with rhodamine-conjugated phalloidin (ThermoFisher Scientific) to visualize cell shape, the cells were dried and mounted with ProLong Diamond Antifade Mountant (Thermo Fisher Scientific). The slides were examined using an SP8 confocal microscope (Leica, Wetzlar, Germany). Images were captured using the Leica LAS-X image acquisition software. Overlays were generated using LAS-X software. In the images of VEGFR2- or Nrf2-immunostained MIO-M1 cells, the levels of immunofluorescence intensity were measured, after normalization to the background, using the “Analysis” menu of Adobe Photoshop (Adobe Photoshop CS3; Adobe Systems, Mountain View, CA, USA).

### 2.2. Ex Vivo Studies

#### 2.2.1. Preparation of Retinal Explants

Ex-vivo studies were performed using 3- to 5-week-old C57BL/6J mice. The procedures were approved by the Commission for Animal Wellbeing of the University of Pisa (permission number: 0034612/2017) and were in compliance with the ARVO Statement for the Use of Animals in Ophthalmic and Vision Research, the Italian guidelines for animal care (DL 26/14), and the EU Directive (2010/63/EU). The mice were kept in a regulated environment (23 ± 1 °C, 50 ± 5% humidity) with a 12 h light/dark cycle (lights on at 8:00 a.m.). Retinas, after dissection in Modified Eagle Medium (MEM, Sigma-Aldrich), were cut into 4 fragments and were transferred onto Millicell-CM culture inserts (Merck Millipore, Darmstadt, Germany) with ganglion cells up. Eight fragments were transferred on each insert. Then, the inserts were placed in 6-well tissue culture plates with 1 mL of medium culture. The culture medium was composed of 50% MEM (Sigma-Aldrich), 25% Hank’s buffer salt solution (HBSS, Sigma-Aldrich), 25% Dulbecco’s Phosphate Buffered Saline (DPBS, Sigma-Aldrich), 25 U/mL penicillin (Sigma-Aldrich), 25 mg/mL streptomycin (Sigma-Aldrich), 1 µg/mL amphotericin B (Sigma-Aldrich), and 200 µM l-glutamine (Sigma-Aldrich). The explants were incubated at 37 °C with 5% CO_2_.

#### 2.2.2. Pharmacological Treatments

Retinal explants were incubated for 24 h with different concentrations of recombinant mouse VEGF (ThermoFisher Scientific), with 100 µM H_2_O_2_ (Sigma-Aldrich), or with 25 µM of the VEGFR2 inhibitor SU1498 (Sigma-Aldrich).

#### 2.2.3. Preparation of Conditioned Media

To produce CM, the retinal explants were incubated in 1 mL of culture medium containing 100 µM H_2_O_2_ for 24 h (CM-24h) or 5 days (CM-5D). In the preparation of CM-5D the culture medium was changed every day, therefore the CM was the medium of the fifth day of incubation. The CMs were collected and centrifuged for 5 min at 1200× *g* to remove cell debris and 900 µL of supernatant was transferred to fresh retinal explants that were cultured and treated for 24 h with CM-24h or CM-5D.

#### 2.2.4. Quantitative Real-Time PCR

Total RNA was extracted using RNeasy Mini Kit (Qiagen, Hilden, Germany), resuspended in RNase-free water and quantified by spectrophotometric analysis (BioSpectrometer, Eppendorf). First-strand cDNA was generated from 1 µg of total RNA using the QuantiTect Reverse Transcription Kit (Qiagen). The qPCR analysis was carried out using Sso Advanced Universal SYBR Green Supermix (Bio-Rad Laboratories) on a CFX Connect Real-Time PCR System and CFX manager software (Bio-Rad Laboratories). The primers were designed according to published mouse cDNA sequences in the GenBank database: *VEGF* 5′-GCACATAGGAGAGATGAGCTTCC-3′ forward and 5′CTGCGCTCTGAACAAGGCT-3′reverse; *rpl13a* 5′- CACTCTGGAGGAGAAACGGAAGG-3′ forward and 5′-GCAGGCATGAGGCAAACAGTC-3′reverse. mRNA was quantified with the ΔΔCt method; *VEGF* mRNA levels were normalized to *rpl13a* mRNA levels as endogenous control.

#### 2.2.5. ELISA

ELISA was performed as described for the in vitro studies using a kit (R&D Systems) for mouse VEGF. After statistical analysis, data from the different experiments were plotted and averaged in the same graph.

#### 2.2.6. Immunofluorescence

Mouse retinal fragments were fixed in 4% paraformaldehyde in 0.1M phosphate buffer for 2 h and then stored overnight at 4 °C in 25% sucrose in 0.1M phosphate buffer. Subsequently, they were embedded in cryo-gel, frozen using liquid nitrogen and then cut into 10-μm-thick coronal sections with a cryostat. The sections were mounted onto gelatinized slides and stored at −20 °C until use. For immunostaining, the sections were incubated overnight with antibodies directed to same anti-VEGFR2 antibody used for the in vitro studies (1:400 dilution) and then with an anti-rabbit secondary antibody conjugated with Alexa-Fluor-488 (Life Technologies, 1:200 dilution) at room temperature for 2 h. The slides were coverslipped with Fluoroshield Mounting Medium containing DAPI (Abcam). The images were acquired using an epifluorescence microscope (Nikon Europe, Amsterdam, The Netherlands) and adjusted for contrast and brightness using Adobe Photoshop. The levels of immunofluorescence intensity were measured, after normalization to the background, using the “Analysis” menu of Adobe Photoshop.

### 2.3. Statistics

Statistical significance was evaluated using t-test, one-way ANOVA followed by Newman–Keuls, or uncorrected Fisher’s LSD multiple comparison post-hoc tests. The results were expressed as mean ± SEM of the indicated n values (Prism 6; GraphPad software, San Diego, CA, USA). Differences with *p* < 0.05 were considered significant.

## 3. Results

### 3.1. OS Induces VEGFR2-Dependent VEGF Expression Both in MIO-M1 Cells and in Retinal Explants

First, we wanted to evaluate the effects of OS on *VEGF* expression after 24 h treatment both in MIO-M1 cells and in retinal explants. To determine the concentration of H_2_O_2_ needed to induce OS in MIO-M1 cells without significantly affecting cell viability, a test of cell proliferation/vitality was performed in the presence of different doses of H_2_O_2_. As shown in Appendix A, MIO-M1 cells maintained a level of proliferation/vitality and a morphology similar to controls up to an H_2_O_2_ concentration of 400 µM, while higher concentrations resulted in dramatic loss of cell viability and altered cell morphology. Therefore, OS in MIO-M1 cells was induced using 400 µM H_2_O_2_. Concerning retinal explants, we have shown previously that 5-day incubation in OS (100 µM H_2_O_2_) induces a significant increase of *VEGF* mRNA expression [4]. Here, to determine whether a 24 h incubation in OS is a sufficient time to detect an effect of OS in retinal explants, the expression of the transcription factor *Nrf2* and of phase II antioxidant enzymes NAD(P)H quinone dehydrogenase 1 (*NQO1*), superoxide dismutase (*SOD*), and glutamate-cysteine ligase catalytic subunit (*GCLC*) was measured. As shown in Appendix A, the expression of *Nrf2*, *NQO1*, and *SOD* mRNA became significantly different from control values after 24 h incubation, while *GCLC* mRNA appeared to increase during the first 24 h incubation without reaching significant difference from control values. Together, these data indicate that, in our model of retinal explants, OS effects can be appreciated after 24 h incubation.

Both in MIO-M1 cells (Figure 1A) and in retinal explants (Figure 1B), *VEGF* mRNA expression was significantly increased after 24 h incubation in OS. In addition, to ascertain whether the increased *VEGF* mRNA expression induced by OS was mediated by VEGFR2, the experiments were replicated in the presence of a VEGFR2 inhibitor. The data showed that blockade of VEGFR2 with 0.1 µM Apatinib completely inhibited the OS-induced increase of *VEGF* mRNA expression in MIO-M1 cells (Figure 1A). The VEGFR2 blockade with 25 µM SU1498 induced the same effect in retinal explants (Figure 1B).

### 3.2. Exogenous VEGF Induces VEGFR2-Dependent VEGF Expression Both in MIO-M1 Cells and in Retinal Explants

To ascertain whether VEGF can induce *VEGF* expression in Müller cells, MIO-M1 cells were incubated for 24 h in the presence of various concentrations of recombinant exogenous VEGF (exo-VEGF). Significant increases of *VEGF* mRNA expression were observed with exo-VEGF at 1 and 5 ng/mL, but not at 10 ng/mL (Figure 1C). Similar to MIO-M1 cells, retinal explants incubated for 24 h in the presence of various concentrations of exo-VEGF also displayed an increase of *VEGF* mRNA expression. In particular, significant effects were obtained with 1 or 10 ng/mL of exo-VEGF, while no changes were observed with the dose of 100 ng/mL (Figure 1D).

These increases of *VEGF* mRNA expression in the presence of exo-VEGF were likely to be mediated by activation of VEGFR2. Indeed, VEGFR2 blockade with Apatinib in MIO-M1 cells or with SU1498 in retinal explants completely inhibited the effects of 1 ng/mL or 10 ng/mL, respectively, of exo-VEGF (Figure 1E,F). In addition, in immunofluorescence experiments, VEGFR2 immunostaining (Figure 2A,B) was remarkably increased both in MIO-M1 cells and in retinal explants following treatment with the most effective dose of exo-VEGF (Figure 2C,D), while this increase was inhibited by blockade of VEGFR2 with Apatinib (MIO-M1 cells; Figure 2E) or with SU1498 (retinal explants; Figure 2F). A quantitative analysis of VEGFR2 immunofluorescence levels confirmed the statistical significance of these changes (Figure 2G,H).

Concerning the localization of VEGFR2 in MIO-M1 cells, punctate VEGFR2 immunostaining appeared to be distributed mainly at the cell border, consistent with VEGFR2 localization to the plasma membrane of MIO-M1 cells (Figure 2A,E). In the presence of exo-VEGF, some VEGFR2 immunolabeling was also seen inside the cells (Figure 2C), consistent with a possible VEGFR2 internalization following binding with exo-VEGF. In control retinal explants, VEGFR2 immunoreactivity was detected in profiles resembling blood vessels, mainly localized to the outer plexiform layer (OPL), and in a few putative Müller cell processes (Figure 2B). In contrast, in explants incubated for 24 h with 10 ng/mL exo-VEGF, immunolabeled blood vessels were seen in high amounts not only in the OPL but also in the inner plexiform layer and in the ganglion cell layer, while intensely labeled Müller cell processes could be seen spanning the whole retinal thickness (Figure 2D).

### 3.3. Conditioned Medium Induces VEGF Expression in MIO-M1 Cells

The overexpression of *VEGF* mRNA induced by OS in MIO-M1 cells was concomitant with an increase of VEGF release in the culture medium (Figure 3A). In particular, we recorded a circa 3-fold increase of the amount of released VEGF, ranging from 181.7 ± 33.2 pg/mL in controls to 566.4 ± 63.1 pg/mL in OS-treated MIO-M1 cells. The CM from these cultures (CM-OS) and CM from cultures of untreated MIO-M1 cells (CM-Ctrl) were used to incubate newly plated MIO-M1 cells. As shown in Figure 3B, the levels of *VEGF* mRNA showed a 5- to 6-fold increase in cells incubated in CM-OS with respect to untreated cells and an about 3-fold increase with respect to cells treated with CM-Ctrl. This effect was significantly inhibited by the VEGFR2 blocker Apatinib. Interestingly, CM-Ctrl resulted in a 2.5-fold increase of *VEGF* mRNA expression with respect to untreated cells, while in this case Apatinib was ineffective. Although this increase did not result in statistical significance with ANOVA followed by Newman–Keuls post-test, it gained significance when the data were analyzed with uncorrected Fisher’s LSD test. This observation indicates that *VEGF* mRNA expression in newly plated MIO-M1 cells may be affected also by some additional factors other than VEGF that are present in CM-Ctrl. In immunofluorescence experiments (Figure 3C–G), VEGFR2 immunostaining was remarkably increased in MIO-M1 cells incubated in CM-OS, where, as in the case of exo-VEGF treatment, it appeared to localize within the cells (Figure 3F). This increase of VEGFR2 immunoreactivity was inhibited by treatment with Apatinib (Figure 3G).

### 3.4. Conditioned Medium Does Not Induce VEGF Expression in Retinal Explants

Consistent with our previous studies [4], 24 h OS induced a significant increase in VEGF release from retinal explants, which reached a concentration of about 1.4 pg/mL (Figure 4A). However, the CM from these cultures, that is CM-24h, did not induce any significant increase of *VEGF* mRNA expression in fresh retinal explants (Figure 4B). We considered the possibility that 24 h OS was a time too short to allow the release of a sufficient amount of VEGF to induce *VEGF* mRNA expression in other explants, therefore we repeated the experiments using a CM-5D, that is the medium collected on the fifth day of incubation in OS. As shown in Figure 4C, VEGF release in these conditions was considerably increased and the CM-5D contained 5.22 ± 0.87 pg/mL of VEGF; however, similar to CM-24h, it did not induce any significant *VEGF* mRNA upregulation when used as medium for culturing fresh retinal explants, although a tendency toward an increased *VEGF* mRNA expression could be noted (Figure 4D). This result was likely to be due to excessive dilution of the released VEGF in the CM (see discussion), indicating that the model of retinal explants has some limits that do not allow further investigation of the possible effects of VEGF expressed and released in response to OS. Therefore, the study was continued using only the MIO-M1 cell line as an experimental model.

### 3.5. OS Triggers Nrf2 Nuclear Translocation, while OS-Induced VEGF Expression Is Inhibited by Nrf2 Blockade

OS is known to induce Nrf2 translocation to the nucleus, where it promotes the expression of antioxidant enzymes [22,23,24]. Our hypothesis is that this mechanism also leads to *VEGF* expression. Therefore, as a first step, we verified that OS promotes Nrf2 nuclear translocation in our model, and we found that treatment of MIO-M1 cells with 400 µM H_2_O_2_ for 24 h led to a significant increase of Nrf2 immunofluorescence, both in the cytoplasm and in the cell nuclei, an effect that was inhibited by concomitant treatment of MIO-M1 cell cultures with the Nrf2 inhibitor ML385 (Figure 5A–D). Normalization of the Nrf2 immunofluorescence intensity in the nucleus to the total intensity (nucleus + cytoplasm, not shown) indicated that ML385 induced a general decrease of Nrf2 production. In addition, Nrf2 blockade with 5 µM ML385 abolished the increase of *VEGF* mRNA expression induced by OS (Figure 5E). Together, these data show that OS-induced *VEGF* expression is mediated by Nrf2 activation and nuclear translocation.

### 3.6. Exo-VEGF Has No Effect on Nrf2 Nuclear Translocation, while Exo-VEGF-Induced VEGF Expression Is Not Affected by Nrf2 Blockade

If OS triggers VEGF expression and release through a Nrf2-mediated mechanism, as shown by the evidence described above, the autocrine loop inducing further *VEGF* expression is likely to be mediated by a different mechanism. Therefore, we tested whether exo-VEGF might induce Nrf2 nuclear translocation and whether Nrf2 blockade might affect exo-VEGF-induced *VEGF* expression. The results showed that the treatment of MIO-M1 cells with exo-VEGF did not induce any changes in nuclear Nrf2, as assessed with immunofluorescence analysis (Figure 6A–D), while Nrf2 blockade did not influence the increase of *VEGF* mRNA expression induced by treatment with exo-VEGF (Figure 6E). Together, these data show that *VEGF* expression induced by exo-VEGF is not Nrf2-dependent.

### 3.7. Blockade of HIF-1 Inhibits VEGF mRNA Expression in All Conditions

As expected, a functional HIF-1 is necessary for *VEGF* expression. Indeed, as shown in Figure 7, the increases of *VEGF* mRNA induced by either OS or exo-VEGF were prevented by ACF, a HIF-1 inhibitor, at 5 µM concentration. Of note, the treatment of MIO-M1 cells with ACF alone also determined a significant decrease of *VEGF* mRNA levels with respect to the control, untreated MIO-M1.

## 4. Discussion

VEGF is a widely known growth factor whose levels increase with the progression of DR, causing a functional shift from neuroprotective to pro-angiogenic effects. In the adult retina, *VEGF* is expressed in neurons, endothelial cells, astrocytes, and Müller cells [20,25], and its expression and release are regulated by a series of mediators, including receptors, kinase proteins, and transcription factors [26,27,28]. In addition, VEGF is also known to act in an autocrine manner to regulate its own expression in some circumstances [15,16,17,18,19]. Here we have shown that in the retina VEGF upregulation may be determined by an autocrine loop and that this mechanism can be activated by an adaptive response to OS, a pathologic condition typical of DR. This conclusion is supported by the results obtained in this study, which can be summarized as follows: i) both OS and exo-VEGF induce a VEGFR2-mediated increase of *VEGF* mRNA expression; ii) in both cases, this increase is blocked by an inhibitor of HIF-1 dimerization, indicating that HIF-1 acts as a sort of final common effector for *VEGF* expression; iii) OS causes VEGF release, indicating that the increased *VEGF* mRNA expression is followed by VEGF production and release; iv) the VEGF released in response to OS triggers a VEGFR2-mediated increase of *VEGF* mRNA expression; v) the OS-induced increase of *VEGF* mRNA is mediated by Nrf2 nuclear translocation; vi) the exo-VEGF-induced increase of *VEGF* mRNA does not depend on Nrf2 activation. Together, these data allow a hypothetical reconstruction of a (simplified) mechanism explaining how VEGF expression and release are induced and sustained in a condition of OS similar to the one that is likely to affect the retina in DR (Figure 8).

### 4.1. The Role of OS

A large body of experimental evidence demonstrates that different types of stress may provoke VEGF expression and release in the retina. In particular, a variety of factors and conditions induced in the retina by diabetes, ranging from inflammation to advanced glycation end-products to excitotoxicity, or to OS, are likely to result in increased VEGF production [3,26,29,30]. Among these factors, OS seems to play a prominent role as a major inducer of retinal VEGF, and here we demonstrate that OS can cause VEGF expression and release in sufficient amounts to trigger an autocrine loop leading to further VEGF release. In particular, considering that treatments with antioxidant compounds reduce the OS-induced VEGF upregulation in models of DR [6], the possibility exists that OS not only induces but also is involved in sustaining the VEGF autocrine loop. A major intracellular mediator involved in this mechanism is Nrf2, since its inhibition blocks the OS-induced increase of *VEGF* mRNA expression as demonstrated by the present results. Nrf2 is a transcription factor that regulates the expression of different antioxidant genes and whose expression is tightly regulated. In normal conditions, Nrf2 is constitutively targeted by Keap1, a cytoskeletal protein able to sequester Nrf2 in the cytoplasm and to destine it to proteasomal degradation. In the presence of OS, the Nrf2-Keap1 interaction is resolved, and both free and newly synthesized Nrf2 translocates into the nucleus, where it stimulates the expression of a plethora of antioxidant genes by binding to the antioxidant response elements (AREs) in their promoters [31]. As discussed below, there is evidence indicating that a Nrf2-regulated pathway may also influence VEGF levels [32].

In this regard, it should be noted that the Nrf2 blocker that we used in these studies, ML385, does not interfere with Nrf2 nuclear translocation, but blocks Nrf2 transcriptional activity by interfering with binding of Nrf2 to the ARE in the promoter region of Nrf2 target genes [33]. On the other hand, it has been reported that Nrf2 binding to ARE not only promotes antioxidant gene expression, but also triggers a positive feedback loop leading to further *Nrf2* expression [34,35]. Therefore, our immunofluorescence observations, summarized in Figure 5, may reflect the fact that OS results both in increased Nrf2 nuclear translocation and in increased *Nrf2* expression, while treatment with ML385 inhibits Nrf2-triggered *Nrf2* expression, thereby reducing significantly the amount of Nrf2 that may translocate to the nucleus.

### 4.2. The Involvement of VEGFR2

Our data show that both MIO-M1 cells and retinal explants increase *VEGF* mRNA expression in response to OS or to exo-VEGF. In addition, the observation that VEGFR2 blockade prevents the effect of both OS and exo-VEGF indicates that if OS is sufficient to trigger VEGF release (pathway “1” in Figure 8), an appreciable increase of VEGF expression needs an activation of VEGFR2 (pathway “2” in Figure 8). Interestingly, our immunofluorescence data suggest that exo-VEGF may increase VEGFR2 production in MIO-M1 cells and in retinal Müller cells (see Figure 2). Similar observations have been reported in bovine aorta endothelial cells [9] and indicate the existence of an autocrine VEGF loop that also involves VEGFR2 production. There is experimental evidence in endothelial cells that VEGF-induced VEGFR2 upregulation may be induced through a NF-κB-dependent pathway [9,36].

Regarding the dose-response relationship between the dose of exo-VEGF and the amount of *VEGF* mRNA expression, we have noted that for doses of exo-VEGF higher than 5 ng/mL for MIO-M1 cells and 10 ng/mL for retinal explants, the effect is reduced and the *VEGF* mRNA levels become similar to those in control conditions. This decreased *VEGF* mRNA expression in the presence of increasing exo-VEGF doses is likely to be due to desensitization caused by VEGFR2 internalization, as suggested by our immunofluorescence data in MIO-M1 cells and by observations of VEGF-driven VEGFR2 internalization in different experimental models [37,38,39,40].

### 4.3. The Involvement of Müller Cells

Müller cells constitute the main glial cell type in the retina, where they play a variety of fundamental functions, including establishment and maintenance of the blood–retina barrier, light conduction through the retinal thickness, recycling of the retinal chromophore, clearance of extracellular glutamate, release of antioxidants (e.g., glutathione), glycogen storage, electrolytic balance, water clearance regulation, and neuronal survival [41]. Müller cells are also likely to play a primary role in the VEGF-driven autocrine loop described in this study. Indeed, as shown by our immunofluorescence data, they seem to be the only retinal cell type, together with endothelial cells, displaying evident VEGFR2 expression, and both MIO-M1 cells and Müller cells in retinal explants greatly increase VEGFR2 expression in response to exo-VEGF. Müller cells are the main source of VEGF in the retina and they release VEGF in pathologic conditions such as DR [42,43,44]. In addition, VEGF released by Müller cells is protective for the Müller cells themselves and for retinal neurons [21,22]. In particular, the observation reported here that ACF significantly decreases *VEGF* mRNA expression in untreated MIO-M1 cells suggests that Müller cells are likely to express and release basal levels of VEGF. This VEGF production may support an autocrine signaling that seems to be essential for the maintenance of Müller cells themselves, since significant increases of apoptosis and autophagy have been reported in Müller cells as a consequence of systemic neutralization of VEGF [12] or of bevacizumab administrations [21]. Consistent with this view, conditional VEGFR2 deletion in Müller cells was found to induce a significant decrease of Müller cell density in the retinas of diabetic mice [43,44].

In our experiments with MIO-M1 cells, CM-OS, similar to exo-VEGF, increased both VEGFR2 immunofluorescence and *VEGF* mRNA expression. Interestingly, the amount of VEGF in the CM-OS was in the range of the doses of exo-VEGF that induced *VEGF* mRNA expression. Inhibition of VEGFR2 in MIO-M1 cells prevented these effects, thus demonstrating that they were due to VEGF, although our observations indicate that MIO-M1 cells may also release some other factors that could contribute to stimulate *VEGF* expression. For instance, it is known that inflammatory cytokines may promote *VEGF* expression [45].

These effects of the CM-OS could not be observed in retinal explants, probably due to limitations of the model. Indeed, the VEGF released in the CM-OS on the fifth day of incubation by 8 explants (corresponding to 2 retinas) was, on average, 5.22 pg diluted in 1 mL of culture medium. This demonstrated an excessive dilution of the VEGF in the culture medium resulting in about 200-fold lower than the minimal dose of exo-VEGF being able to trigger *VEGF* expression (1 ng/mL, see Figure 1B). Although the volume of the medium as well as the accumulation period could be theoretically adjusted to reach effective concentrations of VEGF in the CM-OS, variations in these parameters could result in deleterious effects on the tissue viability and the reproducibility of the experimental procedures. However, considering the amount of VEGF released in the volume of the mouse vitreous calculated in 4–5 µL [46], it follows that, in the presence of OS, 0.52–0.65 ng/mL of VEGF would be released into a mouse eye, a dose near the one of exo-VEGF that is able to elicit an increase of *VEGF* mRNA expression in retinal explants. These considerations encourage further investigations in vivo using, for instance, intravitreal injections of exo-VEGF or vitreous fluid from diabetic animals.

Together, our observations indicate that retinal Müller cells may react to OS expressing and releasing an amount of VEGF sufficient to stimulate VEGFR2 and to trigger activation of an autocrine VEGF loop (Figure 8). Based on our immunofluorescence data, the possibility exists that endothelial cells, which also express VEGFR2, may also contribute to the VEGF autocrine loop triggered by OS, and their potential involvement will be assessed in dedicated studies. Finally, retinal pigment epithelial cells represent an additional retinal cell type that may be involved in these mechanisms since they have been observed to respond to OS with an increased *VEGF* expression [13].

### 4.4. Intracellular Pathways

Our results are consistent with the summary diagram depicted in Figure 8. Both “pathway 1” (triggered by OS) and “pathway 2” (triggered by VEGF) converge onto HIF-1, as demonstrated by the observation that both OS- and exo-VEGF-induced *VEGF* expression are blocked by HIF-1 inhibition. In particular, the HIF-1 inhibitor used in the present studies, ACF, is an inhibitor of HIF-1 dimerization. Indeed, HIF-1 comprises a labile α subunit and a stable β subunit. HIF-1α is rapidly hydroxylated and degraded through proteasomal machinery at normal oxygen tension, while in hypoxia it escapes degradation and translocates to the nucleus, where it associates with HIF-1β. The resulting heterodimeric HIF-1 triggers the expression of a large number of genes by binding to hypoxia response elements (HREs) in their promoter regions, which function as cis-acting elements that determine transcriptional activation of HIF-1 target genes. Among them, the *VEGF* gene harbors a HRE in its promoter and its expression is strictly controlled by HIF-1 [47].

In addition to oxygen, the stability of HIF-1α may be regulated by a variety of other factors, including growth factors, cytokines, hormones, reactive oxygen species, microRNAs, lncRNAs, and intracellular effectors [48,49,50,51,52]. In particular, in accordance with “pathway 1” of Figure 8, a variety of experimental observations indicate that OS may stimulate HIF-1α stabilization through Nrf2 activation. Indeed, evidence has been obtained in cancer cells that Nrf2 may directly regulate the expression of *HIF1A*, the gene encoding HIF-1α [53], while in cardiomyocytes HIF-1α upregulation has been reported to be related to Nrf2 activation through heme oxygenase-1 (HO-1) expression [54]. The involvement of HO-1 as a mediator of Nrf2 influence on HIF-1α has been confirmed by the observation that HO-1 metabolites, including CO and bilirubin, increased HIF-1α stability in astrocytes [55]. In addition, the possibility has been proposed that Nrf2-regulated HO-1 expression may contribute to a positive feedback loop in which VEGF activates Nrf2 in an ERK1/2-dependent manner and the consequent HO-1 upregulation would increase VEGF expression, likely through CO production and HIF-1α stabilization [32,56], although our observations reporting no effects of Nrf2 blockade after exo-VEGF administration do not seem to support this possibility.

Concerning the evidence supporting the existence of an autocrine mechanism shown as “pathway 2” in Figure 8, our data demonstrate that exo-VEGF induces *VEGF* expression independent from Nrf2, thus indicating that “pathway 2” may function independently from “pathway 1” in the presence of adequate VEGF levels. Results obtained in different types of endothelial cells suggest that VEGF-stimulated VEGFR2 could stimulate HIF-1α stabilization through activation of signal transducer and activator of transcription (STAT) 3 [15,16]. In general, STAT3 is considered an intracellular effector of central importance for VEGF-mediated angiogenesis. Indeed, in endothelial cells STAT3 has been found to be implicated in the transduction of VEGF signals [19], while in different types of tumors activation of STAT3 has been reported to increase VEGF expression [57].

## 5. Conclusions

This study supports the existence of a VEGF autocrine loop that can be triggered by OS in the retina and that may be of central importance for the onset and development of DR. Moreover, our observations point at Müller cells being the main retinal cell type involved in these mechanisms, although a possible contribution of the endothelial cells of the retinal microvasculature or of the retinal pigment epithelium cannot be excluded. These findings derive from evaluations of in vitro and ex vivo models, which, despite some limitations, allowed us to highlight the existence of this mechanism through a controlled modulation of the VEGF pathway. The outcome of these studies confirms the importance of OS in the early phases of DR and provides a rationale that can be further examined in in vivo studies. Most importantly, it emphasizes the notion that DR could be prevented, or at least attenuated, with the simple administration of antioxidant substances, many of which can be found in natural foods [58,59], to diabetic patients.

## Figures and Tables

**Figure 1 cells-09-01452-f001:**
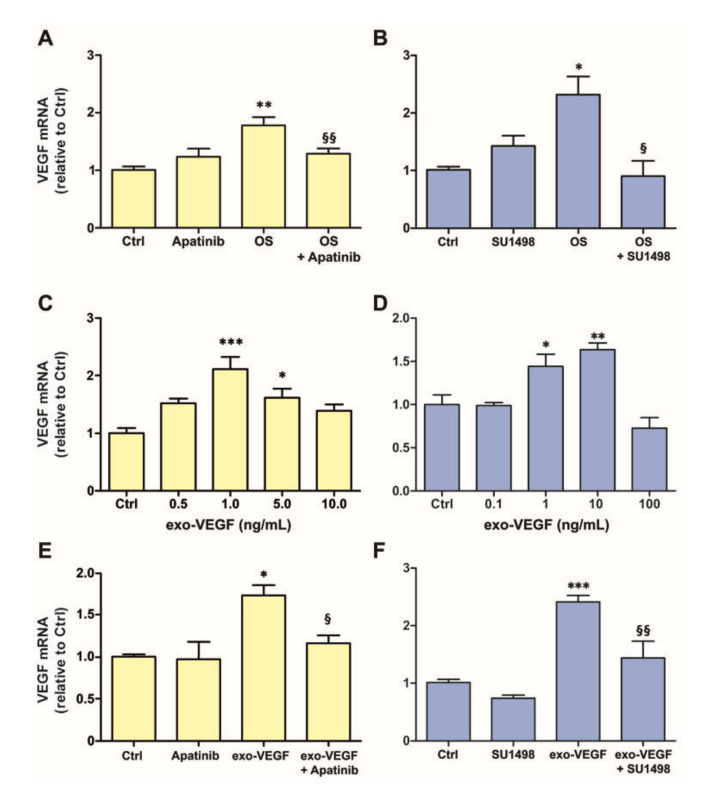
(**A**) and (**B**) represent vascular endothelial growth factor (*VEGF*) mRNA expression in MIO-M1 cells and in retinal explants, respectively, exposed to oxidative stress (OS; 400 µM H_2_O_2_ for MIO-M1 cells and 100 µM H_2_O_2_ for retinal explants) for 24 h and effect of a vascular endothelial growth factor receptor 2 (VEGFR2) blocker (0.1 µM Apatinib for MIO-M1 cells and 25 µM SU1498 for retinal explants). * *p* < 0.05 and ** *p* < 0.01 relative to controls (Ctrl); ^§^
*p* < 0.05 and ^§§^
*p* < 0.01 relative to OS. (**C**) and (**D**) represent *VEGF* mRNA expression in MIO-M1 cells and in retinal explants, respectively, exposed to different concentrations of exogenous VEGF (exo-VEGF) for 24 h. * *p* < 0.05 ** *p* < 0.01 and *** *p* < 0.001 relative to Ctrl. (**E**) and (**F**) represent *VEGF* mRNA expression in MIO-M1 cells and in retinal explants, respectively, in response to the most effective concentration of exo-VEGF (1 ng/mL in MIO-M1 cells and 10 ng/mL in retinal explants) and the effect of a VEGFR2 blocker (Apatinib for MIO-M1 cells and SU1498 for retinal explants). * *p* < 0.05 and *** *p* < 0.001 relative to Ctrl; ^§^
*p* < 0.05 and ^§§^
*p* < 0.01 relative to exo-VEGF. *n* = 3 in (**A**–**E**); *n* = 5 in (**F**).

**Figure 2 cells-09-01452-f002:**
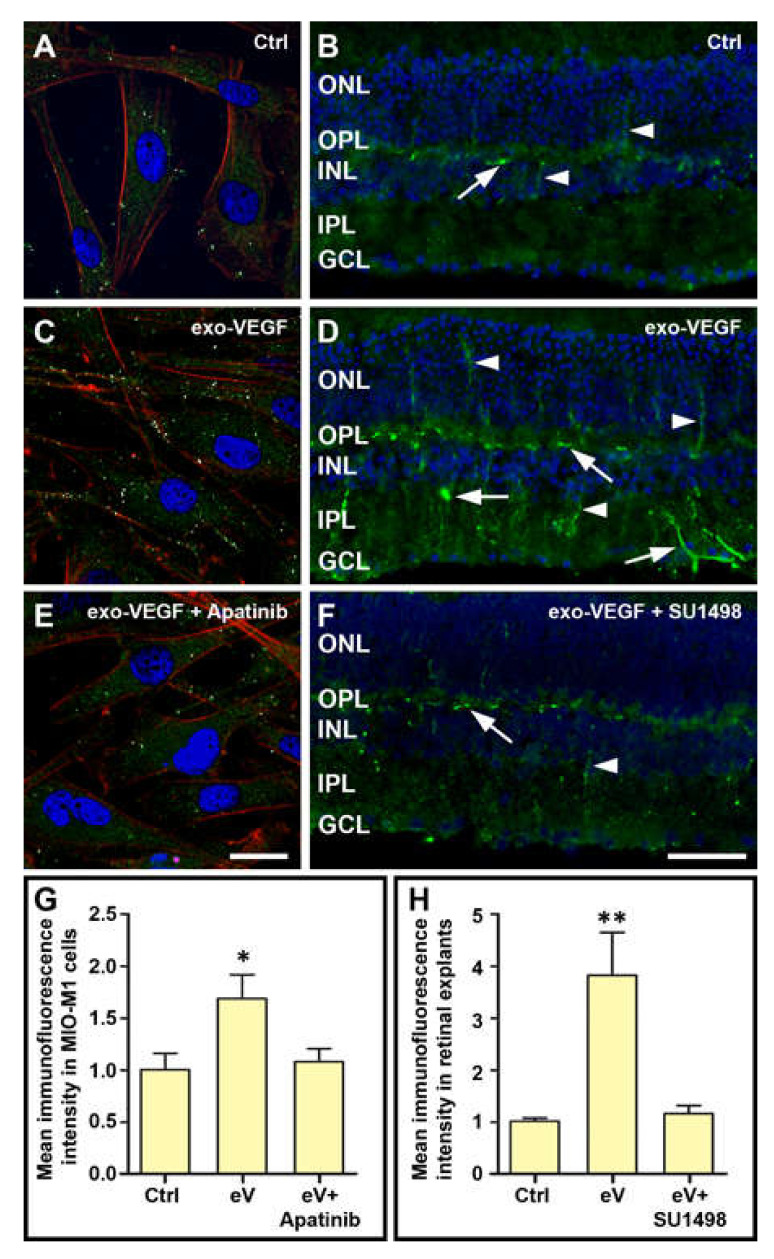
VEGFR2 immunofluorescence in MIO-M1 cells (**A**,**C**,**E**) and in cryostat sections of retinal explants (**B**,**D**,**F**) in Ctrl (**A**,**B**), in the presence of 1 ng/mL (MIO-M1 cells) or 10 ng/mL (retinal explants) of exo-VEGF (**C**,**D**) and in the presence of exo-VEGF with a VEGR2 blocker (0.1 µM Apatinib for MIO-M1 cells and 25 µM SU1498 for retinal explants: **E**,**F**). In the images of MIO-M1 cells (**A**,**C**,**E**), above background, specific VEGFR2 immunolabeling was highlighted using Adobe Photoshop, and it appears as bright, whitish dots. Background, non-specific immunostaining is dark green. Cell nuclei were visualized with Hoechst and actin filaments with rhodamine-conjugated phalloidin. In retinal explants (**B**,**D**,**F**), specific VEGR2 immunostaining is bright green, while cell nuclei were visualized with DAPI counterstain. Putative immunolabeled blood vessels are indicated by arrows; putative immunolabeled Müller cell processes are indicated by arrowheads. GCL, ganglion cell layer; INL, inner nuclear layer; IPL, inner plexiform layer; ONL, outer nuclear layer; OPL, outer plexiform layer. Scale bars, 20 µm. **G** and **H** indicate VEGFR2 immunofluorescence levels normalized to Ctrl in MIO-M1 cells and in retinal explants, respectively, in the different experimental conditions. eV, exo-VEGF. * *p* < 0.05 and ** *p* < 0.01 relative to Ctrl. *n* = 4.

**Figure 3 cells-09-01452-f003:**
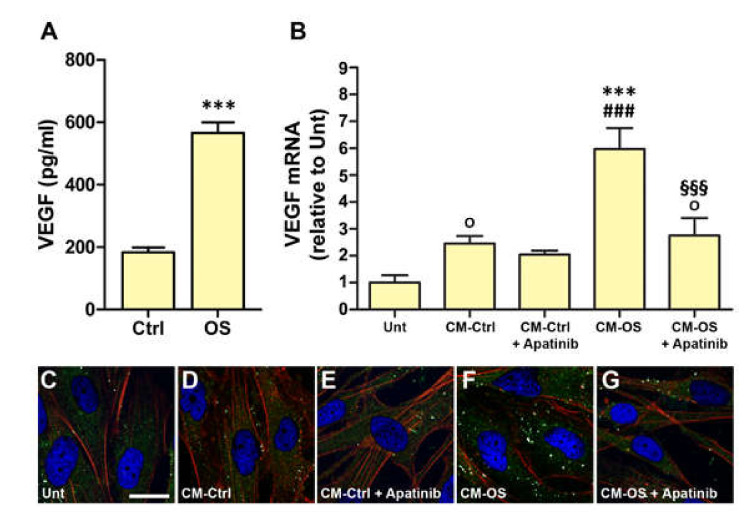
(**A**) VEGF release in the culture medium of MIO-M1 cells exposed to OS for 24 h, as measured with ELISA. (**B**) *VEGF* mRNA expression in MIO-M1 cells incubated for 24 h in the conditioned medium of untreated MIO-M1 cells (CM-Ctrl) or of MIO-M1 cells exposed to OS for 24 h (CM-OS), with or without the VEGFR2 blocker Apatinib at 0.1 µM. *** *p* < 0.001 relative to control MIO-M1 cell cultures (Ctrl) (**A**) or to untreated MIO-M1 cell cultures (that is incubated in non-conditioned medium, Unt) (**B**), ^§§§^
*p* < 0.001 relative to CM-OS, and ^###^
*p* < 0.001 relative to CM-Ctrl as evaluated with one-way ANOVA followed by Newman–Keuls post-hoc test. ^o^
*p* < 0.005 with respect to Unt as evaluated with one-way ANOVA followed by uncorrected Fisher’s LSD post-hoc test. *n* = 4 in (**A**); *n* = 6 in (**B**). (**C**–**G**): Images of VEGFR2 immunofluorescence in MIO-M1 cells in the experimental conditions as in (**B**). Other details as in Figure 2. Scale bar, 20 µm.

**Figure 4 cells-09-01452-f004:**
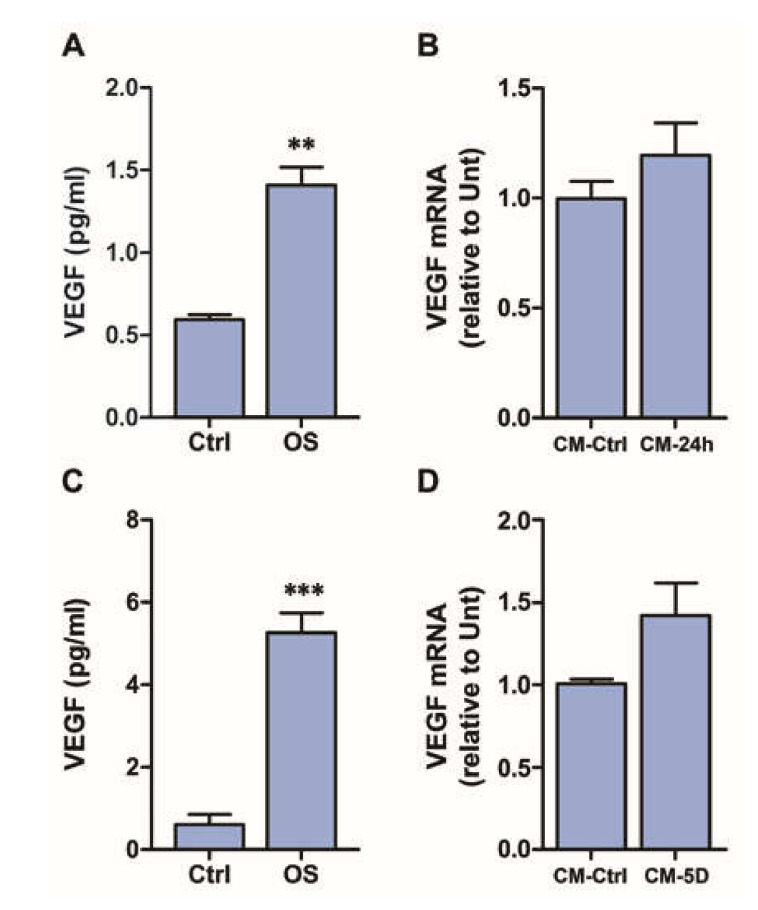
(**A**) VEGF release in the culture medium of retinal explants exposed to OS for 24 h, as measured with ELISA. (**B**) *VEGF* mRNA expression in retinal explants incubated for 24 h in non-conditioned medium (Ctrl) or in explants incubated for 24 h in the conditioned medium of explants exposed to OS for 24 h (CM-24h). (**C**) VEGF release in the culture medium of retinal explants exposed to OS for 5 days, as measured with ELISA. (**D**) *VEGF* mRNA expression in retinal explants incubated for 24 h in the conditioned medium of untreated explants (CM-Ctrl) or in explants incubated for 24 h in the conditioned medium of explants exposed to OS for 5 days (CM-5D). ** *p* < 0.01 and *** *p* < 0.001 relative to Ctrl. *n* = 3 in all graphs.

**Figure 5 cells-09-01452-f005:**
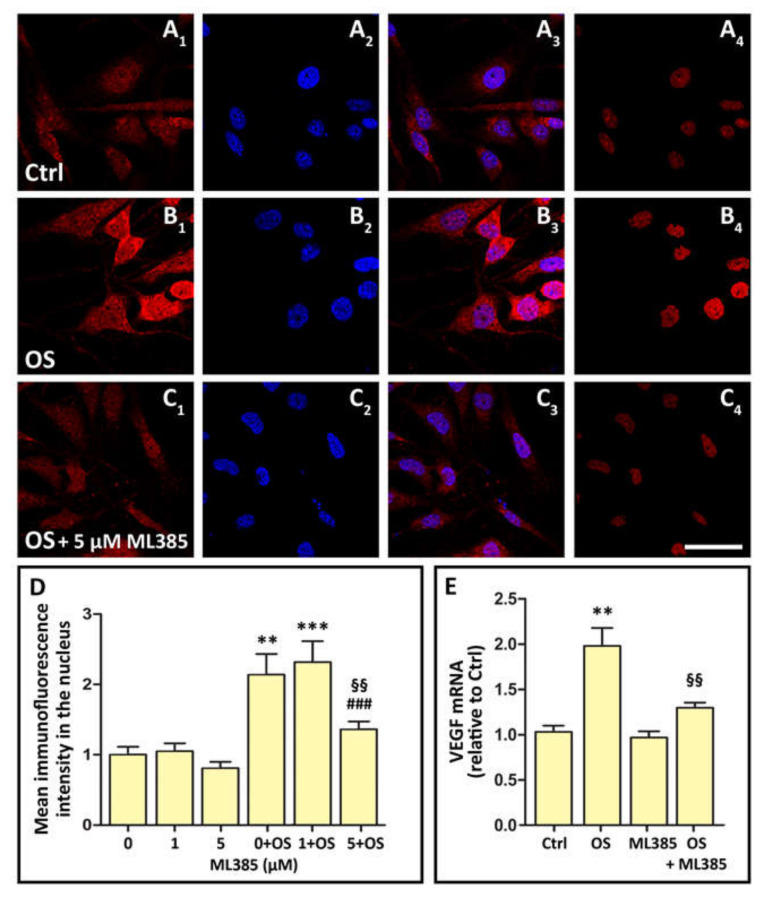
Immunofluorescence images documenting nuclear factor erythroid 2-related factor 2 (Nrf2) production and nuclear translocation in Ctrl, OS-treated (OS) and in OS-treated MIO-M1 cells incubated in the presence of the Nrf2 inhibitor ML385 at 5 µM (OS + ML385). (**A1**), (**B1**) and (**C1**) show the overall Nrf2 immunostaining of MIO-M1 cells in the three different conditions; (**A2**), (**B2**) and (**C2**) show the Hoechst-stained cell nuclei; (**A3**), (**B3**) and (**C3**) are merged images of the previous two; (**A4**), (**B4**) and (**C4**) show Nrf2 immunostaining limited to the cell nucleus. Scale bar, 50 µm. The histograms in (**D**) represent the quantification of immunofluorescence intensity within the nuclei of MIO-M1 cells in the different experimental conditions. The values are relative to that corresponding to 0 µM ML385. ** *p* < 0.01 and *** *p* < 0.001 relative to 0 µM ML385; ^§§^
*p* < 0.01 relative to OS + 0 µM ML385; ^###^
*p* < 0.001 relative to OS + 1 µM ML385. The histograms in (**E**) represent *VEGF* mRNA expression in MIO-M1 cells exposed to OS for 24 h and effect of the Nrf2 inhibitor ML385. ** *p* < 0.01 relative to Ctrl; ^§§^
*p* < 0.01 relative to OS. *n* = 8 in (**D**); *n* = 3 in (**E**).

**Figure 6 cells-09-01452-f006:**
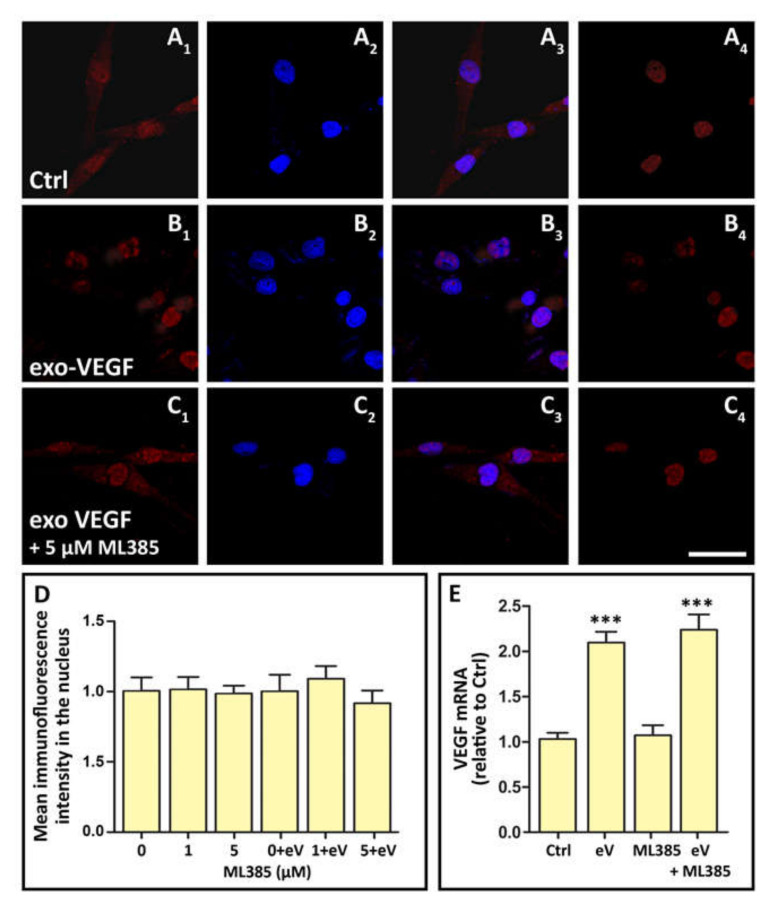
Immunofluorescence images documenting Nrf2 production and nuclear translocation in Ctrl, in exo-VEGF-treated and in exo-VEGF-treated MIO-M1 cells incubated in the presence of the Nrf2 inhibitor ML385 (exo-VEGF + ML385). (**A1**), (**B1**) and (**C1**) show the overall Nrf2 immunostaining of MIO-M1 cells in the three different conditions; (**A2**), (**B2**) and (**C2**) show the Hoechst-stained cell nuclei; (**A3**), (**B3**) and (**C3**) are merged images of the previous two; (**A4**), (**B4**) and (**C4**) show Nrf2 immunostaining limited to the cell nucleus. Scale bar, 50 µm. The histograms in (**D**) represent the quantification of immunofluorescence intensity within the nuclei of MIO-M1 cells in the different experimental conditions. The values are relative to that corresponding to 0 µM ML385; eV, exo-VEGF. The histograms in (**E**) represent *VEGF* mRNA expression in MIO-M1 cells exposed to exo-VEGF (eV) for 24 h and effect of the Nrf2 inhibitor ML385. *** *p* < 0.001 relative to Ctrl. *n* = 4 in (**D**); *n* = 3 in (**E**).

**Figure 7 cells-09-01452-f007:**
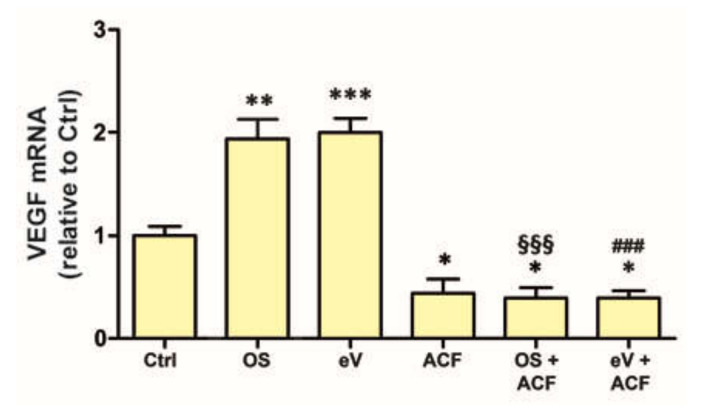
*VEGF* mRNA expression in MIO-M1 cells exposed to OS or to exo-VEGF (eV) for 24 h and effect of the hypoxia inducible factor-1 inhibitor acriflavine (ACF) at 5 µM. * *p* < 0.05, ** *p* < 0.01 and *** *p* < 0.001 relative to Ctrl; ^§§§^
*p* < 0.001 relative to OS; ^###^
*p* < 0.001 relative to eV. *n* = 4.

**Figure 8 cells-09-01452-f008:**
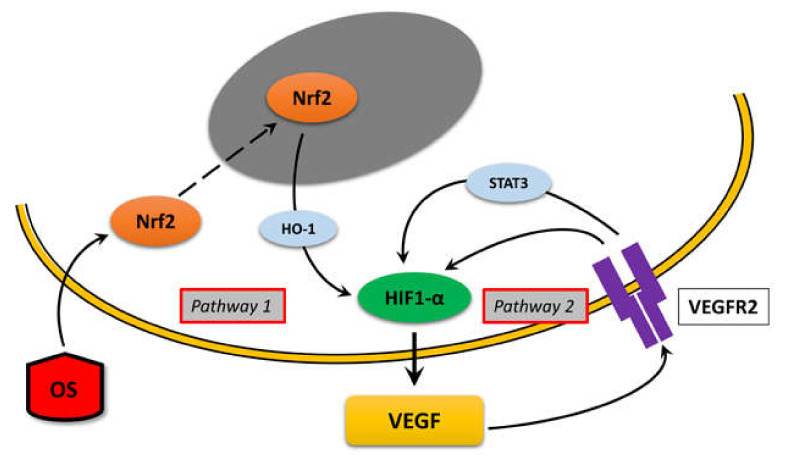
Schematic interpretation of the results of the present study and of other data in the literature showing the possible mechanism of an OS-induced VEGF autocrine loop in the retina. This mechanism can be divided into two parts (pathways). According to our observations, in “pathway 1”, OS triggers Nrf2 activation and nuclear translocation. In the nucleus, Nrf2 induces the expression of antioxidant genes, and in particular the one coding the HO-1 enzyme, which is reported in the literature to be able to stabilize HIF-1α. For the sake of simplicity, the diagram does not indicate HIF-1α or STAT3 nuclear translocation, induction of *VEGF* transcription, VEGF translation, or VEGF release, but simply indicates that HIF-1α stabilization results in VEGF release. This activates “pathway 2”, in which the released VEGF would bind to VEGFR2, which in turn (likely through STAT3 activation, according to the literature) would again stimulate HIF-1α stabilization and nuclear translocation for further VEGF expression and release.

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
