# Peer review of "Oxidative Stress Induces a VEGF Autocrine Loop in the Retina: Relevance for Diabetic Retinopathy"

_cells, 2020, doi:10.3390/cells9061452_

Round 1

Reviewer 1 Report

COMMENTS FOR THE AUTHOR:

            The present study by Rossino and colleagues reports a set of experimental findings supporting a VEGF autocrine loop in Muller cells. The study shows that oxidative stress and exo-VEGF increase VEGF mRNA and VEGFR2 content in MIO-M1 cells and retinal explants Muller cells in a VEGFR2 dependent way. The increase in VEGF mRNA expression also involves Nrf2 since it is partially prevented by Nrf2 inhibition. The study also demonstrates that oxidative stress increases VEGF release in both culture MIO-M1 and retinal explants. Additionally, oxidative stress, but not exo-VEGF, promotes the increase in Nrf2 immunoreactivity intensity overall in cells and in nuclei, which is blockage by Nrf2 inhibition. Finally, the increase induced by oxidative stress and VEGF in MIO-M1 cells is completely prevented by HIF1-α inhibitor.   The manuscript is well written and clear.  The topic is clinically relevant since oxidative stress is a factor involved in a wide range of retinal pathologies, such as diabetic retinopathy and glaucoma.

I have some general comments and more specific issues that the authors may wish to address.

  • In general, the Materials and Methods has sufficient description. However, some details may be added:
  1. Culture cell density of MIO-M1 is not mentioned.
  2. To facilitate the readers, it would be kind to add time of drug treatment (VEGF, H2O2, ML385, Apatinib and acriflavine). It is mentioned in the results, but not in the methods.
  3. Concerning immunofluorescence in MIO-M1 cells, in my experience the time of fixation with paraformaldehyde and the detergent (and its concentration) is quite problematic. Especially the use of 0.3% of Triton X-100 in cell culture can affect cell morphology and, therefore, immunolabelling. Although, the Nrf2 data seem similar to what have been shown to this protein, the pattern of immunolabelling of VEGFR2, like small dots, does not follow exactly the pattern in retinal cell cultures. So, maybe it would be interest to show, in a supplementary material, photomicrographs of cells in phase/DIC mode to show the healthy morphology, or at least state that in the results.

  • Results

Figure 1

  1. Concerning the MIO-M1 cells treatment with 400 µM H2O2, it would be interest to have an idea of the morphology of the cells. The data show no cell death but how are the appearance of MIO-M1 cells? Do the cells look healthy or deteriorate?
  2. Maybe it would be more adequate to name concentrations than dose in Fig 1 legend.
  3. Why use different VEGFR2 inhibitors in the experiments showed in figure 1 depending on model (culture or explants).
  4. Finally, what the explanation for the absence of effect in higher concentrations of exo-VEGF (10 ng/mL in MIO-M1 cells and 100 ng/mL in explants)? Is there downregulation of VEGFR2 when cells are exposed to higher concentrations of VEGF? If yes, would it occur in vivo? How do the authors fit the autocrine loop in a downregulation scenario? Maybe a small discussion about this question would be beneficial to the field.

    Concerning the immunofluorescence analysis showing VEGFR2 increase induced by VEGF/OS in MIO-M1 cells, the results would be greatly benefited of a quantitative approach. Maybe the authors could consider to do some western blot experiments or quantify the images.

Figure 4

  1. It appears that the description of CM-5D in methods is different from results, since in the former says that the culture medium was changed every day and in the latter states that CM is from the 5 days culture. Which one is correct?
  2. The authors conclude that increase in VEGF release confirm that VEGF mRNA is being translated. Would the authors test inhibitors of translation and measure VEGF release?
  3. Maybe the more interesting question remaining to be answered in the present study is whether the production/secretion of VEGF depends on HIFα.
  4. Finally, the authors state, in the discussion, that VEGF expression and release are induced and sustained in OS condition. The results from the present study corroborate the “induced”, but it is not clear that this is sustained by the time. Perhaps to affirm that the authors have to perform a temporal analysis of VEGF production/release.

Figure 5

  1. The result that ML385 prevents the increase in nuclear Nrf2 immunodetection. I understand the possibility, discussed by the authors, that Nrf2 decrease in the nucleus could be due to the inhibition by ML385 of Nrf2-induced Nrf2 expression. I was wondering if the normalization of the Nrf2 intensity in the nucleus by the total intensity (nucleus + cytoplasm) could help to verify whether it is related to a general decrease in Nrf2 production or it is related to nuclear content. In this matter, it is known that nuclear content can be modulated by modulation of Nrf2 nuclear exclusion as well.
  2. In retinal pigment epithelium cells, OS activates VEGFR2 and phosphor-Akt. One target of Akt is GSK3-β, which in turn can regulate Nrf2 nuclear exclusion. Then, maybe it would be interesting to discuss a bit this possibility.

Figure 8

In retinal endothelial cells, the activation of VEGFR2 induces the translocation of STAT3 to the nucleus, which stimulates VEGF expression. Therefore, maybe it would be nice to have this possibility depicted in the scheme.

            Finally, have all my solidarity concerning the difficult period we are living with covid-19 pandemic. The authors have performed a wonderful study with an adequate design and important results. The suggestions made is just some possible ideas/thoughts to add in case the authors agreed, and mainly, if the authors have the conditions to work safely at the lab.

            Therefore, in my opinion the paper is in the scope of Cells and acceptable with minor revisions.

Author Response

The present study by Rossino and colleagues reports a set of experimental findings supporting a VEGF autocrine loop in Muller cells. The study shows that oxidative stress and exo-VEGF increase VEGF mRNA and VEGFR2 content in MIO-M1 cells and retinal explants Muller cells in a VEGFR2 dependent way. The increase in VEGF mRNA expression also involves Nrf2 since it is partially prevented by Nrf2 inhibition. The study also demonstrates that oxidative stress increases VEGF release in both culture MIO-M1 and retinal explants. Additionally, oxidative stress, but not exo-VEGF, promotes the increase in Nrf2 immunoreactivity intensity overall in cells and in nuclei, which is blockage by Nrf2 inhibition. Finally, the increase induced by oxidative stress and VEGF in MIO-M1 cells is completely prevented by HIF1-α inhibitor.   The manuscript is well written and clear.  The topic is clinically relevant since oxidative stress is a factor involved in a wide range of retinal pathologies, such as diabetic retinopathy and glaucoma.

I have some general comments and more specific issues that the authors may wish to address.

In general, the Materials and Methods has sufficient description. However, some details may be added:

Culture cell density of MIO-M1 is not mentioned.

R – This information has been added (lines 84-85).

To facilitate the readers, it would be kind to add time of drug treatment (VEGF, H2O2, ML385, Apatinib and acriflavine). It is mentioned in the results, but not in the methods.

R – This information has been added (lines 102 and 164).

Concerning immunofluorescence in MIO-M1 cells, in my experience the time of fixation with paraformaldehyde and the detergent (and its concentration) is quite problematic. Especially the use of 0.3% of Triton X-100 in cell culture can affect cell morphology and, therefore, immunolabelling.

Although, the Nrf2 data seem similar to what have been shown to this protein, the pattern of immunolabelling of VEGFR2, like small dots, does not follow exactly the pattern in retinal cell cultures. So, maybe it would be interest to show, in a supplementary material, photomicrographs of cells in phase/DIC mode to show the healthy morphology, or at least state that in the results.

R – A supplementary figure (new Supplementary figure 1) has been added to show the morphology of MIO-M1 cells before and after fixation plus permeabilization using DIC optics. Some text has also been added in the methods section (lines 130-131).

Results

Figure 1

Concerning the MIO-M1 cells treatment with 400 µM H2O2, it would be interest to have an idea of the morphology of the cells. The data show no cell death but how are the appearance of MIO-M1 cells? Do the cells look healthy or deteriorate?

R – Images of MIO-M1 cells in DIC microscopy have been added to Supplementary figure 2 to document that MIO-M1 cells display healthy morphology when treated with up to 400 µM H2O2. A comment on cell morphology has been added in the Results section (lines 213 and 215).

Maybe it would be more adequate to name concentrations than dose in Fig 1 legend.

R – These changes have been made in the legend of figure 1.

Why use different VEGFR2 inhibitors in the experiments showed in figure 1 depending on model (culture or explants).

R – The experiments on MIO-M1 cells and those on retinal explants were performed in two distinct laboratories, one located at the University of Florence, the other at the University of Pisa. The two teams had performed previous, independent preliminary experiments using VEGFR2 inhibitors, obtaining reliable results with Apatinib in MIO-M1 cells and with SU1498 in retinal explants. In addition, when we tried to treat MIO-M1 cells with SU1498, we obtained inconsistent results indicating that, in our culture conditions and with our experimental protocols, SU1498 was not functioning as a reliable VEGFR2 inhibitor for MIO-M1 cells. Therefore, we decided to use two different VEGFR2 inhibitors to obtain replicable conditions in MIO-M1 cells on the one hand and in retinal explants on the other. We think this is too complicated to explain in the manuscript.

Finally, what the explanation for the absence of effect in higher concentrations of exo-VEGF (10 ng/mL in MIO-M1 cells and 100 ng/mL in explants)? Is there downregulation of VEGFR2 when cells are exposed to higher concentrations of VEGF? If yes, would it occur in vivo? How do the authors fit the autocrine loop in a downregulation scenario? Maybe a small discussion about this question would be beneficial to the field.

R – As discussed in paragraph 4.2 (lines 469-475), we interpret this reduction of VEGF expression in the presence of high concentrations of exo-VEGF with the possibility that the high amount of ligand induces a high rate of VEGFR2 downregulation due to receptor internalization. This hypothesis is supported by the cited literature. We do not think that this could occur in vivo because the highest exo-VEGF concentrations used in our experiments are far beyond the highest possible amounts of released VEGF in normal eyes or even in eyes subjected to high levels of stress. Indeed, we have shown that MIO-M1 cells or retinal explants challenged with oxidative stress respond with a release of VEGF that is comparable to that inducing further VEGF release (see lines 497-498 and 511-514) and very far from that inducing VEGFR2 desensitization. Therefore, a VEGF autocrine loop would not have to face a downregulation scenario neither in normal nor in pathologic conditions. We think the discussion of this point in paragraph 4.2 is sufficient, and adding further hypotheses and considerations would unnecessarily increase the length of the discussion.

Concerning the immunofluorescence analysis showing VEGFR2 increase induced by VEGF/OS in MIO-M1 cells, the results would be greatly benefited of a quantitative approach. Maybe the authors could consider to do some western blot experiments or quantify the images.

R – We agree with this observation of the Reviewer. Indeed, although the increase of VEGFR2 immunofluorescence is apparent in the images of figure 2, immunofluorescence (and, in general, immunohistochemistry) is not a quantitative technique and statements about changes in the levels of the antigen should be substantiated by other approaches or by further analyses of the immunofluorescent material. In the present case, we are not able to perform Western blots, at least in the period of 10 days allowed for the revision of this paper, due to the restrictions imposed by the covid-19 pandemic. However, we could use the acquired immunofluorescence images to perform an analysis of the fluorescence levels using a method that we have also used in previous studies. These data, obtained both in MIO-M1 cells and in retinal explants, have been summarized in the new panels G and H of figure 2. The text has been changed accordingly in the legend of figure 2, in the Methods (lines 143-144 and 200-201) and in the Results (lines 261-262).

Figure 4

It appears that the description of CM-5D in methods is different from results, since in the former says that the culture medium was changed every day and in the latter states that CM is from the 5 days culture. Which one is correct?

R – The text in the Results has been changed to clarify that CM-5D is the medium collected on the fifth day of incubation in OS (lines 323 – 324).

The authors conclude that increase in VEGF release confirm that VEGF mRNA is being translated. Would the authors test inhibitors of translation and measure VEGF release?

R – In the present study, VEGF release has been evaluated to get an indication that VEGF mRNA was translated and we did not use inhibitors of translation because this was beyond the scope of the paper. However, we thank the reviewer for this good suggestion that we will take into account for our future work.

Maybe the more interesting question remaining to be answered in the present study is whether the production/secretion of VEGF depends on HIFα.

R – This is a correct observation. In effect, our data demonstrate that both VEGF mRNA expression and VEGF release are induced by OS, while HIF-1α inhibition abolishes VEGF mRNA expression. Together, these observations indicate, although they do not demonstrate directly, that VEGF gene transcription promoted by HIF-1α is followed by VEGF protein translation and VEGF release.  It has been widely established in the literature that the stabilization of HIF-1α occurring under several stress conditions, including OS, increases VEGF production by promoting VEGF gene expression (Kurihara et al., Adv Exp Med Biol 2014, 801:275-81; Bonello et al., Arterioscler Thromb Vasc Biol 2007, 27:755-61). Moreover, the HIF-mediated upregulation of VEGF has been shown to increase protein levels and secretion of VEGF. In particular, HIF-1α stabilization with dimethyloxallyl glycine or its inhibition with YC-1 in retinal explants cultured under hypoxic stress have been shown to directly correlate with VEGF protein content and VEGF secretion in the culture medium (Mei et al., J Pathol 2012, 226:519-33). Similarly, blockade of HIF-1 α with digoxin in hypoxic MIO-M1 cells inhibited VEGF mRNA expression and protein secretion (Xin et al., PNAS 2013, 110: E3425-E3434). In the present work, we demonstrated that HIF-1α is involved in the VEGF autocrine loop as a final common effector for both the OS and the VEGF/VEGFR2 signaling. Considering the evidence in the literature based on both retinal explants and MIO-M1 models, our observations strongly indicate, although they do not demonstrate, that HIF-1α-induced VEGF mRNA expression results in an increase of VEGF protein content in the tissue and of VEGF release in the medium. We thought that discussing this point would have “overloaded” the discussion.

Finally, the authors state, in the discussion, that VEGF expression and release are induced and sustained in OS condition. The results from the present study corroborate the “induced”, but it is not clear that this is sustained by the time. Perhaps to affirm that the authors have to perform a temporal analysis of VEGF production/release.

R – There is wide evidence that, in the retina, OS causes VEGF upregulation. Moreover, studies evaluating the efficacy of antioxidants (for instance natural compounds such as polyphenols, carotenoids and others – See Rossino and Casini, 2019, Nutrients, 2019, 11) demonstrated that these compounds counteract OS-induced VEGF upregulation. In our opinion, these results strongly suggest that OS not only induces, but also contributes to sustain the VEGF autocrine loop. Indeed, if OS was involved in triggering but not in sustaining the VEGF autocrine loop, treatments with antioxidant compounds, which in in vivo models of DR last of 8 - 12 weeks at the least, should not be able to reduce VEGF upregulation. Therefore, we agree with the Reviewer that we used the term “sustain” in an inappropriate way. For this reason, we tried to briefly expose our point of view in the discussion (lines 424-426), and we have replaced the term “sustain” with others more appropriate to the context.

Figure 5

The result that ML385 prevents the increase in nuclear Nrf2 immunodetection. I understand the possibility, discussed by the authors, that Nrf2 decrease in the nucleus could be due to the inhibition by ML385 of Nrf2-induced Nrf2 expression. I was wondering if the normalization of the Nrf2 intensity in the nucleus by the total intensity (nucleus + cytoplasm) could help to verify whether it is related to a general decrease in Nrf2 production or it is related to nuclear content.

R – Thank you for this suggestion. We did the analysis proposed by the Reviewer and found that oxidative stress induced a comparable increase of Nrf2 immunofluorescence both in the cytoplasm and in the nucleus of MIO-M1 cells. Similarly, the treatment with ML385 uniformly decreased Nrf2 immunofluorescence throughout the cell. Therefore, we concluded that the effect of ML385 was a general decrease in Nrf2 production. A brief description of this analysis has been added in the Results (lines 348-350).

In this matter, it is known that nuclear content can be modulated by modulation of Nrf2 nuclear exclusion as well. In retinal pigment epithelium cells, OS activates VEGFR2 and phosphor-Akt. One target of Akt is GSK3-β, which in turn can regulate Nrf2 nuclear exclusion. Then, maybe it would be interesting to discuss a bit this possibility.

R – This is a quite interesting, although complex, issue. As previously demonstrated by Byeon et al. (IOVS 2010, 51:1190-1197) hydrogen peroxide-induced oxidative stress stimulates the release of VEGF by RPE cells. VEGF, in turn, would act as an autocrine factor activating the VEGFR2/PI3K/Akt pathway to enhance RPE survival. In addition, it is also known that in RPE cells Akt activates GSK3-β (see for instance Baek et al., Mol. Vis. 2016, 22:1015-1023) and that GSK3-β phosphorylates Nrf2 to induce its nuclear exclusion and degradation (see for Ref. Culbreth and Alschner, F1000Research 2018, 7:1043). To the best of our knowledge, an effect of GSK3-β inducing Nrf2 nuclear exclusion has not been demonstrated in RPE or Müller cells, and the possibility that the VEGF/VEGFR2 axis may interfere with the nuclear levels of Nrf2 is an interesting aspect that deserves to be developed in further studies, as it may add to the complexity of the connections between Pathway 1 and Pathway 2 described in Figure 8. However, we prefer to avoid introducing this further point in the Discussion for several reasons: i. the mechanism by which oxidative stress induces VEGF production in RPE cells was not investigated by Byeon et al., therefore the possibility exists that, according to our data in MIO-M1 cells, oxidative stress increases VEGF production through Nrf2 activation (Pathway 1 of figure 8) and not necessarily through VEGFR2 activation; ii. as explained in the discussion (lines 438-441), ML385 is known to reduce the transcription of the Nrf2 gene without interfering with Nrf2 nuclear translocation, as also confirmed by our new observations on Nrf2 immunolabeling (see previous point), therefore it seems unlikely that nuclear exclusion is involved in our model, and in any case we do not have enough “matter” to speculate further; iii. there are multiple intracellular pathways modulating nuclear Nrf2 that are activated by VEGF, including for instance the downstream ERK1/2 (Kweider et al., JBC 2011, 286: 42863-42872), and all would deserve attention; iv. as demonstrated by the experiments depicted in Figure 6, the addition of exogenous VEGF to MIO-M1 cells does not affect nuclear levels of Nrf2, thus suggesting that, in these conditions, both the mechanisms leading to Nrf2 nuclear exclusion (as those activated by GSK3-β) and those inducing Nrf2 nuclear translocation (as those activated by ERK1/2) are activated at the same time and therefore compensate each other. Alternatively, as mentioned on lines 548-549, we must admit that in our model there are no detectable effects of VEGF-induced VEGF-R2 activation on Nrf2 modulation.

Figure 8

In retinal endothelial cells, the activation of VEGFR2 induces the translocation of STAT3 to the nucleus, which stimulates VEGF expression. Therefore, maybe it would be nice to have this possibility depicted in the scheme.

R – The schematic reconstruction of figure 8 is purposely very simple because we wanted to concentrate on the existence, supported by our data, of two different but interacting pathways (indicated as Pathway 1 and Pathway 2) that are likely to play a fundamental role in determining the VEGF levels in the early phases of DR. Similar to the nuclear translocation of STAT3, also that of HIF-1α has been omitted, as explained in the figure legend, and other downstream pathways originating from VEGFR2 activation have not been shown. In a previous publication of one of the present Authors (Dal Monte et al., Naunyn Schmiedebergs Arch Pharmacol 2011, 383, 593-612), a similar but much more detailed scheme was provided. In this case, we privileged simplicity to deliver, in the clearest possible way, the main message deriving from our study. In the legend, we have added that the diagram does not indicate STAT3 nuclear translocation.

Finally, have all my solidarity concerning the difficult period we are living with covid-19 pandemic. The authors have performed a wonderful study with an adequate design and important results. The suggestions made is just some possible ideas/thoughts to add in case the authors agreed, and mainly, if the authors have the conditions to work safely at the lab.

Therefore, in my opinion the paper is in the scope of Cells and acceptable with minor revisions.

R – We thank the Reviewer for his (her) solidarity and we look forward to better times for all of us.

Reviewer 2 Report

The paper of Rossino et al. clearly describes the concept of the VEGF positive autofeedback loop through VEGF2R. The experimental design is good, the execution is also excellent. I particularly like the summary drawing, it is helpful to understand the message of the paper. 

The only limitation that I detected during this study is that as authors themselves state  "...the model of retinal explantshas some limits that do not allow investigation further the possible effects of VEGF expressed and released in response to OS."

Although I understand this statement, I have expected some further explanation in the Discussion. This issue should be discussed, as well as the potential ways to prove this hypothesis in in vivo model(s).

Another issue for further discussion is why exogenously applied VEGF did not have an effect on Nrf2 function. What is a possible cause of this discrepancy?

After adding these two short paragraphs to the discussion, the paper will be in good shape for publication.

Author Response

The paper of Rossino et al. clearly describes the concept of the VEGF positive autofeedback loop through VEGF2R. The experimental design is good, the execution is also excellent. I particularly like the summary drawing, it is helpful to understand the message of the paper.

The only limitation that I detected during this study is that as authors themselves state  "...the model of retinal explants has some limits that do not allow investigation further the possible effects of VEGF expressed and released in response to OS." Although I understand this statement, I have expected some further explanation in the Discussion. This issue should be discussed, as well as the potential ways to prove this hypothesis in in vivo model(s).

R – We agree with the limitations of the retinal explant model highlighted by the Reviewer. In effect, the main limitation of this model consisted in the culture parameters, such as the medium volume and the medium replacement frequency, which influenced the efficacy of the CM-OS. In particular, considering the amount of VEGF released in OS, we concluded that the excessive dilution of the VEGF occurring in the CM did not allow the latter to be effective in inducing VEGF expression once administered to non-stressed explants. However, nor the medium volume or the accumulation period can be modified without avoiding alteration of the tissue viability and the reliability of the model. We rearranged the text (lines 503-514) in order to clarify this point. However, we still wanted to point out that, considering the effects of the exo-VEGF on the retinal explants, the amount of VEGF detected in the medium following the OS condition is compatible with an expected response if released in the volume of a mouse vitreous. Hence, this encourages further investigations using in vivo models. In agreement with the Reviewer comment, we modified the text (lines 514-515) in order to describe the possibility to use intravitreal injection of exo-VEGF or of vitreous fluid from diabetic animals, as parallel approaches in line with this study, to prove our hypothesis in vivo.

Another issue for further discussion is why exogenously applied VEGF did not have an effect on Nrf2 function. What is a possible cause of this discrepancy?

R – In effect, we do not see this result as a discrepancy. Indeed, this observation demonstrates that the autocrine loop (“pathway 2” in figure 8), when activated, does not need, in principle, the activation of the Nrf2-controlled pathway, that is “pathway 1”. In other words, this result supports the existence of the VEGF autocrine loop. Thanks to this comment of the Reviewer, we realized that this aspect was not stressed adequately, therefore we added a few lines (lines 550-553) to better explain the significance of the experiments with exo-VEGF in the presence of the Nrf2 blocker ML385.

After adding these two short paragraphs to the discussion, the paper will be in good shape for publication.

Reviewer 3 Report

In this paper, the authors tested the hypothesis that OS-stimulated VEGF sustains its own expression with an autocrine mechanism. By analyzing the endogenous VEGF expression in vitro and in retinal explants under the treatment of H2O2 and exogenous VEGF and VEGFR2 inhibitors, they tried to investigate the relationship between oxidative stress Nrf2, HIF-1, and VEGF expression. They concluded that there is a retinal VEGF autocrine loop triggered by oxidative stress. The authors claim that this novel mechanism may be of importance in regulating VEGF levels during the development of DR. This is an interesting study. However, I would like to know how this mechanism links to DR other that regulate the VEGF level; and any evidence of stimulation in other cell types, e.g. RPE since the breakdown of outer blood-retina barrier could also be involved in DR.

The results of this study are consistent with the proposed hypothesis. Overall, it can be an interesting attempt for the field. However, images and figures can be improved for clarity by labeling the significant regions on the image by the arrows. The data presentation is overwhelming and can be reorganized by moving some data into supplemental materials. It would be better if the experiment also includes other retinal cell types and different oxidative stimulants, e.g. paraquat or oxLDL.

The following specific points are also for the authors to consider:

The introduction gave a clear background for the experiment, but the rationale of the study can be explained more clearly. Why only Müller cells may also play an important role in the autocrine loop, in which autocrine VEGF binds to VEGF Receptor 2. Additionally, please explain clearly how OS induce Nrf2 activation and nuclear translocation HIF-1 is related in the regulation of VEGF expression.

Experimental design and methods are appropriate.

  • Need to specify why MIO-M1, a spontaneously immortalized Müller cell line is used.
  • Why use retinal explants instead of in vivo injection? Please specify.
  • Studies with in vivo intravitreal injection of recombinant VEGF and other inhibitors may be needed to confirm the findings.
  • For figure 1, please explain why Apatinib increases VEGF expression?
  • Figure 2, expression of VEGFR2 needs to be pointed out with arrows.
  • Figure4, it’s difficult to tell whether OS led to increased Nrf2 expression of translocation, please use arrows to specify

Author Response

In this paper, the authors tested the hypothesis that OS-stimulated VEGF sustains its own expression with an autocrine mechanism. By analyzing the endogenous VEGF expression in vitro and in retinal explants under the treatment of H2O2 and exogenous VEGF and VEGFR2 inhibitors, they tried to investigate the relationship between oxidative stress Nrf2, HIF-1, and VEGF expression. They concluded that there is a retinal VEGF autocrine loop triggered by oxidative stress. The authors claim that this novel mechanism may be of importance in regulating VEGF levels during the development of DR. This is an interesting study. However, I would like to know how this mechanism links to DR other that regulate the VEGF level; and any evidence of stimulation in other cell types, e.g. RPE since the breakdown of outer blood-retina barrier could also be involved in DR.

R – Our aim was to verify the hypothesis that, under stress conditions like those characterizing the early phases of DR, an increased production of VEGF may be sustained by VEGF itself in an autocrine manner. We used a simplified approach involving retinal explants and MIO-M1 cells as experimental models. In particular, we choose MIO-M1 cells because they are known to produce VEGF, to increase VEGF production under stress conditions, and to express VEGFR2. In our opinion, this hypothesis, sustained by the results presented here, is relevant to DR since the activation of such autocrine loop may represent an important molecular switch leading from the early to the late phases of DR, in which oBRB and iBRB breakdown as well as angiogenic processes lay the ground to phenomena finally leading to visual loss. As to the role possibly played in this loop by other retinal cell types that are affected in DR, we have discussed the possibility that endothelial cells may participate (lines 518-520). In addition, we agree with the Reviewer that RPE cells may be an additional candidate, as we have acknowledged with the addition of a few lines in the Discussion (lines 520-522 and 563-564), although the fact that our retinal explants do not include the RPE makes it difficult to speculate about the effective involvement of these cells in an autocrine loop sustaining VEGF production.

The results of this study are consistent with the proposed hypothesis. Overall, it can be an interesting attempt for the field. However, images and figures can be improved for clarity by labeling the significant regions on the image by the arrows. The data presentation is overwhelming and can be reorganized by moving some data into supplemental materials. It would be better if the experiment also includes other retinal cell types and different oxidative stimulants, e.g. paraquat or oxLDL.

R – We agree with the Reviewer that it could be interesting to use other retinal cell types to evaluate cell specificity in response to OS and exo-VEGF treatments. Nevertheless, we believe that our data obtained in Müller cells and retinal explants, which contain all neuroretinal components of the retina, are coherent and adequate to support our hypothesis. In addition, the short time allowed for revising the manuscript and the precarious safety conditions due to covid-19 do not allow extensive additions to the present work. However, the suggestion is definitely positive and we will take it into consideration in planning our work for the near future.

The following specific points are also for the authors to consider:

The introduction gave a clear background for the experiment, but the rationale of the study can be explained more clearly. Why only Müller cells may also play an important role in the autocrine loop, in which autocrine VEGF binds to VEGF Receptor 2.

R – In the Introduction, we hypothesized a major role of Müller cells in the autocrine loop sustaining VEGF expression, but we did not mean that we consider Müller cells as the only retinal cell type involved in this mechanism. We tried to clarify this on lines 60-61. Müller cells are particularly attractive because, as we summarize in the Introduction, they are the main producers of VEGF in the retina, express both VEGFR1 and VEGFR2, depend on VEGF signaling to survive, and signaling of VEGF in Müller cells is necessary to preserve healthy retinal neurons. In our studies, we observed in retinal explants that, in addition to Müller cells, endothelial cells are also likely to increase VEGFR2 expression in response to exo-VEGF treatment, thus probably contributing to the VEGF autocrine loop triggered by OS, a hypothesis supported by the cited literature. In addition, in this revised version of the manuscript, we consider the possibility that retinal pigment epithelial cells may also be involved in this mechanism (lines 520-522 and 563-564).

Additionally, please explain clearly how OS induce Nrf2 activation and nuclear translocation HIF-1 is related in the regulation of VEGF expression.

R – We thank the Reviewer for the suggestion of better clarifying the OS/Nrf2 and HIF-1/VEGF connections, which we think has improved our manuscript. Accordingly, we revised the Discussion giving more precise descriptions of how OS triggers the activation of Nrf2 and of how HIF-1 nuclear translocation activates VEGF transcription (lines 429-435 and 532-535).

Experimental design and methods are appropriate.

Need to specify why MIO-M1, a spontaneously immortalized Müller cell line is used.

R – We used MIO-M1 cells by virtue of their documented morphologic features, marker expression and electrophysiological responses that are comparable to those of primary isolated Müller cells in culture (Limb et al, IOVS 2002, 43:864-869). According to Reviewer’s indication, we added a brief description of the main features of MIO-M1 in the revised manuscript (lines 79-81).

Why use retinal explants instead of in vivo injection? Please specify. Studies with in vivo intravitreal injection of recombinant VEGF and other inhibitors may be needed to confirm the findings.

R – We thank the reviewer for raising this point. The pathophysiology induced by OS as well as the pathways related to VEGF signaling is characterized by a prominent complexity that often limits the effective interaction with and the modulation of these mechanisms in vivo. In effect, typical in vivo variability, systemic interferences, and drug pharmacokinetics could potentially mask mechanistic evidence as to the OS-driven VEGF autocrine loop, affecting the plainness of the outcome. Therefore, despite the simplification of the experimental system, we preferred to begin testing our hypothesis using experimental models that can be better controlled, to build a rationale that can be further examined in vivo. In this context, the role of the retinal explant model is crucial for the translation of the result obtained in vitro to the in vivo condition. Indeed, cultures of retinal explants have been considered among the most suitable models for translational research as they retain the organ structural features and, thus, the complex inter-cellular interactions for the reproduction of in situ phenomena occurring in the neuroretina. Using the ex-vivo approach, we demonstrated that the evidence of the VEGF autocrine loop is not exclusive of the isolated Müller cell culture, but it is confirmed in a higher-complexity system in which this cell type is in intimate contact with other components of the neuroretina. In our opinion, this could represent a step forward in the contextualization of our hypothesis in the organ and encourage future studies in vivo with more consciousness of the mechanisms. Possible future applications of the in vivo models have been added at the line 514-515. In addition, the conclusions paragraph was modified by adding considerations regarding the models.

For figure 1, please explain why Apatinib increases VEGF expression?

R – The Reviewer probably refers to figure 1A, in which the histogram relative to the treatment with Apatinib alone seems to be a little taller than that of controls. However, in all the graphs shown in this study, statistically significant differences from control values are indicated by asterisks (*p<0.05, **p<0.01, ***p<0.001). Since the Apatinib histogram lacks any asterisks, it means that the statistical analysis did not detect any difference in VEGF mRNA expression between control and Apatinib-treated MIO-M1 cells. The same is true for the illusory difference in VEGF mRNA expression between controls and SU1498-treated retinal explants of figure 1F.

Figure 2, expression of VEGFR2 needs to be pointed out with arrows.

R – We thank the Reviewer for this suggestion. Arrows and arrowheads have been added to the photomicrographs of retinal explants in figure 2, and this helps detecting immunolabeled retinal blood vessels and Müller cell processes.

Figure4, it’s difficult to tell whether OS led to increased Nrf2 expression of translocation, please use arrows to specify

R – Probably the Reviewer refers to figure 5. In contrast to the previous point, we think that in this case arrows would not add clarity to the figure. Indeed, panels A4, B4, and C4 depict Nrf2 immunofluorescence limited to the nucleus, and the graph of panel D clearly shows the relative amount of Nrf2 immunofluorescence in cell nuclei in the different experimental conditions.

Reviewer 4 Report

1. Please provide the source of VEGFR2 inhibitor apatinib in the Methos section. Why did the authors use different VEGFR2 inhibitors in the cell and ex vivo expretment?

2. In Figure 1, why did the expression of VEGF mRNA decrease in the higher concentraions of ex-VEGF ?

3. Figure 2, a graph to show the quantitative expression of VEGFR2 will be needed in the immunofluorescence study. In addition, a western blotting expreiment will be better to demonstrated the results.

4. The interaction of VEGF and Nrf2 should be explianed in more detail. The author could refer to the reference of Kweider et al. "Interplay between Vascular Endothelial Growth Factor (VEGF) and Nuclear Factor Erythroid 2-related Factor-2 (Nrf2) :IMPLICATIONS FOR PREECLAMPSIA, THE JOURNAL OF BIOLOGICAL CHEMISTRY VOL. 286, NO. 50, pp. 42863–42872, December 16, 2011".

5. What is the mechanism of VEGF to induce the expression of VEFGR2? The authors did no provide adequate evidence to demonstrate the results. This is the major concerns of this article.

6. Figure 8 is oversimplified. The interaction between VEGF and VEGFR2 should be described in more detail.

Author Response

  1. Please provide the source of VEGFR2 inhibitor apatinib in the Methos section. Why did the authors use different VEGFR2 inhibitors in the cell and ex vivo expretment?

R – The source of Apatinib (Selleck Chemicals, Houston, TX, USA) was indicated on line 98-99. Regarding the two different VEGFR2 inhibitors, as explained above in response to a similar question posed by Reviewer 1, the experiments on MIO-M1 cells and those on retinal explants were performed in two distinct laboratories, one located at the University of Florence, the other at the University of Pisa. The two teams had performed previous, independent preliminary experiments using VEGFR2 inhibitors, obtaining reliable results with Apatinib in MIO-M1 cells and with SU1498 in retinal explants. In addition, when we tried to treat MIO-M1 cells with SU1498, we obtained inconsistent results indicating that, in our culture conditions and with our experimental protocols, SU1498 was not functioning as a reliable VEGFR2 inhibitor for MIO-M1 cells. Therefore, we decided to use two different VEGFR2 inhibitors to obtain replicable conditions in MIO-M1 cells on the one hand and in retinal explants on the other. We think this is too complicated to explain in the manuscript.

  1. In Figure 1, why did the expression of VEGF mRNA decrease in the higher concentraions of ex-VEGF?

R – As discussed in paragraph 4.2 (lines 469-475), we interpret this reduction of VEGF expression in the presence of high concentrations of exo-VEGF with the possibility that the high amount of ligand induces a high rate of VEGFR2 downregulation due to receptor internalization. This hypothesis is supported by the cited literature. As reported above in a response to Reviewer 1, we do not think that this could occur in vivo because the highest exo-VEGF concentrations used in our experiments are far beyond the highest possible amounts of released VEGF in vivo.

  1. Figure 2, a graph to show the quantitative expression of VEGFR2 will be needed in the immunofluorescence study. In addition, a western blotting expreiment will be better to demonstrated the results.

R – We thank the Reviewer for this observation. Indeed, as discussed above in a response to Reviewer 1, immunofluorescence (and, in general, immunohistochemistry) is not a quantitative technique, and statements about changes in the levels of the antigen should be substantiated by other approaches or by further analyses of the immunofluorescent material. In the present case, we are not able to perform Western blots, at least in the period of 10 days allowed for the revision of this paper, due to the restrictions imposed by the covid-19 pandemic. However, we could use the acquired immunofluorescence images to perform an analysis of the fluorescence levels using a method that we have also used in previous studies. These data, obtained both in MIO-M1 cells and in retinal explants, have been summarized in the new panels G and H of figure 2. The text has been changed accordingly in the legend of figure 2, in the Methods (lines 143-144 and 200-201) and in the Results (lines 261-262).

  1. The interaction of VEGF and Nrf2 should be explained in more detail. The author could refer to the reference of Kweider et al. "Interplay between Vascular Endothelial Growth Factor (VEGF) and Nuclear Factor Erythroid 2-related Factor-2 (Nrf2) :IMPLICATIONS FOR PREECLAMPSIA, THE JOURNAL OF BIOLOGICAL CHEMISTRY VOL. 286, NO. 50, pp. 42863–42872, December 16, 2011".

R – In effect, we have referred to this reference (#56 in the reference list) in discussing the possible relationships between VEGF and Nrf2 (lines 545-549). This mechanism deserved some discussion since it is directly related to our results, however its detailed description can be recovered from data in the literature and we thought it was unnecessary to report it here.

  1. What is the mechanism of VEGF to induce the expression of VEFGR2? The authors did no provide adequate evidence to demonstrate the results. This is the major concerns of this article.

R – The Reviewer makes an interesting point. It is true that we did not provide evidence of how VEGF may induce VEGFR2 expression, but this particular issue was not within the scope of the present work. Our aim was to demonstrate the existence of an OS-triggered VEGF autocrine loop in the retina, and the observation that also VEGFR2 seems to be upregulated was not expected. It may be the subject for further analyses, but some evidence is available in the literature. For instance, in endothelial cells the exposure to exo-VEGF has been reported to cause an increase of VEGFR2 expression and of the activation of NF-ĸB in a dose-dependent manner (Gonzalez-Pacheco et al, 2006, reference # 9). In addition, blocking the nuclear translocation of NF-kBp65 has been found to cause a down-regulation of VEGFR2 mRNA and protein expression (Dong et al., 2014, Cancer Biol Ther 15:1479-88). This evidence suggests that a possible mechanism involved in VEGFR2 increase in response to VEGF is mainly driven through a NF-ĸB-dependent pathway. A short note has been added in the discussion to indicate this possibility (lines 455-457).

  1. Figure 8 is oversimplified. The interaction between VEGF and VEGFR2 should be described in more detail.

R – The schematic reconstruction of figure 8 is purposely very simple because we wanted to concentrate on the existence, supported by our data, of two different but interacting pathways (indicated as Pathway 1 and Pathway 2) that are likely to play a fundamental role in determining the VEGF levels in the early phases of DR. As explained in the figure legend, several processes and downstream pathways originating from VEGFR2 activation have been omitted. In a previous publication of one of the present Authors (Dal Monte et al., Naunyn Schmiedebergs Arch Pharmacol 2011, 383, 593-612), a similar but much more detailed scheme was provided. In this case, we privileged simplicity to deliver, in the clearest possible way, the main message deriving from our study. Also in consideration of the appreciation of Reviewer 2 for this figure, we preferred to avoid changes that could distract from the main purpose of this summary diagram.